# The Role of Aesthetics in Intentions to Use Digital Health Interventions

**James L. Denison-Day** [1]*, **Sarah Muir**[1], **Ciaran Newell**[2], **Katherine M. Appleton**[3]

**1** Department of Psychology, University of Southampton, Hampshire, United Kingdom, **2** Dorset Healthcare University NHS Foundation Trust, Poole, United Kingdom, **3** Psychology Department, Bournemouth University, Poole, United Kingdom

* J.L.Denison-Day@soton.ac.uk

**Data Availability Statement:** The data that support the findings of this study are publicly available from the University of Southampton with the identifier https://doi.org/10.5258/SOTON/D2404.

## Abstract

Digital interventions are increasingly recognised as cost-effective treatment solutions for a number of health concerns, but adoption and use of these interventions can be low, affecting outcomes. This research sought to identify how individual aesthetic facets and perceived trust may influence perceptions toward and intentions to use an online health intervention by building on the Technology Acceptance Model, where perceived attractiveness, perceived usefulness, perceived ease of use and perceived enjoyment are thought to predict behavioural intentions towards a website. An online questionnaire study assessed perceptions of nine stimuli varying in four aesthetic facets (simplicity, diversity, colour & craftsmanship), utilising a quasi-experimental within-subjects design with a repetition among three different groups: individuals from the general population who were shown stimuli referring to general health (GP-H) (N = 257); individuals experiencing an eating disorder and shown stimuli referring to eating disorders (ED-ED) (N = 109); and individuals from the general population who were shown stimuli referring to eating disorders (GP-ED) (N = 235). Linear mixed models demonstrated that perceptions of simplicity and craftsmanship significantly influenced perceptions of usefulness, ease of use, enjoyment and trust, which in turn influenced behavioural intentions. This study demonstrates that developing the TAM model to add a further construct of perceived trust could be beneficial for digital health intervention developers. In this study, simplicity and craftsmanship were identified as the aesthetic facets with the greatest impact on user perceptions of digital health interventions.

## Author summary

Digital health interventions are playing a growing role in health care for a wide range of health conditions. However, people do not always engage with these interventions as much as might be hoped for. In this study we looked at a specific factor that might influence the user's first impressions of digital health interventions, and in turn how much they will use them; aesthetic design. By building on an existing model of user acceptance of digital technology, we explored the elements of aesthetic design that are most important in influencing positive judgements when different groups of people access different digital

**Funding:** This work was supported by Bournemouth University as part of a funded PhD supervised by SM, CN and KA, and completed by JDD. The funders had no role in study design, data collection and analysis, decision to publish, or preparation of the manuscript.

**Competing interests:** The authors have declared that no competing interests exist.

health interventions. We found that using designs that look simple and professional improved user's judgements of how useful and trustworthy the intervention is, how easy and enjoyable it is to use, and how likely they would be to use it. This work is intended to help guide future developers to create digital health interventions that are both visually appealing and engaging to users.

## Introduction

Digital health interventions are playing an increasingly important role in the provision of public health, forming a major cornerstone of NHS policy [1]. Whilst evidence indicates that digital health interventions can be a valuable means of addressing a wide range of health conditions [2–4], positive outcomes may be limited, or even entirely negated, if such interventions are unable to attract users and properly engage them with the content. This concern has given rise to increased importance being placed on considering user engagement during intervention development [5–7], with several models and approaches having been developed with this in mind [8–10]. However, whilst these models have marked a definite improvement in the way that intervention developers consider the determinants of user engagement, they often provide only a limited account, or omit entirely, an important design consideration: the visual aesthetics of the intervention.

Visual aesthetics play a role in promoting positive judgements and engagement with digital health interventions [11]. Higher ratings of aesthetics have been found to correlate with higher consumer ratings of digital mental health applications [12]. It has also been suggested that improved website aesthetics can lead to greater levels of website trust in early judgements [13], with perceptions of trust influencing the way in which users evaluate health information websites [14], as well as acting as a predictor of online health activities [15]. Similarly, both design simplicity (presenting content that is well structured) and complexity (presenting content that is varied) have been identified as predictors of perceived ease of use of health information websites [16, 17]. However, it has been suggested that the power of design has not been used to its full potential in the development of web-based interventions [18] and recent research has identified the need for improvement in the aesthetics of digital interventions [19], as well as for evidence-based design strategies to guide their development [20].

Aesthetics can be regarded as an "immediate pleasurable subjective experience that is directed toward an object and not mediated by intervening reasoning" [21]. An aesthetic impression occurs immediately at first sight, rather than being the result of a long lasting cognitive analysis [22]. This places it within the same timeframe as judgements made by users about the website as a whole, as it has been demonstrated that evaluations of online information are made quickly, with viewers judging websites within seconds or even milliseconds [23–25].

The process by which individuals make judgements about digital technologies can be conceptualised using an adapted Technology Acceptance Model (TAM) [26], which follows the well-established causal chain of beliefs + attitude -> intention -> behaviour, known as the Theory of Reasoned Action (TRA; Fishbein & Ajzen, 1975). In this context it has been proposed that aesthetics, or 'perceived attractiveness', may influence users' beliefs, including 'perceived usefulness', 'perceived ease of use' and 'perceived enjoyment', as well as their attitudes towards use of a website, subsequently influencing their intentions and actual usage of the site [26].

Whilst this begins to establish a potential role for aesthetics in promoting user engagement, 'Perceived Attractiveness' fails to offer any distinct insight into which elements of design are most important to consider when developing digital interventions. This might be addressed by

considering developed measures of aesthetics, such as the Visual Aesthetics Website Inventory (VisAWI) developed by Moshagen and Thielsch [21, 27] which assumes four facets of website aesthetics: Simplicity, Diversity, Colourfulness and Craftsmanship.

- **Simplicity** refers to the perceived clarity and structure of the layout of a website.

- **Diversity** refers to the inventiveness and dynamic nature of the layout.

- **Colourfulness** comprises aspects of colour composition, choice and combination.

- **Craftsmanship** refers to the topicality, sophistication and the professionalism of the design.

In addition, it may be important to consider the role of trust in user judgements and intentions towards digital health interventions given the important nature of the information that these websites and applications provide. Indeed, trust has been a known factor in the development of commercial websites, as a result of the impact it has on factors such as purchase intentions and user appeal [13, 28], and has been indicated as playing a role in the selection of online health information [14]. Furthermore, previous research has suggested that improved website aesthetics can lead to greater levels of website trust in early judgements [13, 14], making this a potentially important factor to consider in modelling the role of aesthetic design on user judgements and behavioural intentions.

As such this research sought to identify how individual aesthetic facets may influence intentions and perceptions toward online health interventions, including perceptions of trust, by building on the TAM framework. In order to better understand these mechanisms within the complex and varied context of digital health, this work examined these effects within both the general population and in a population with specific health concerns, eating disorders. Drawing from the elaboration likelihood model (ELM), another important consideration for the design of digital health interventions is the relevance of the content to the user [29]. Content that has higher perceived personal relevance to the user is likely to result in greater levels of engagement, and therefore durable attitude change and consequent behaviour change [30]. Indeed, content relevance has been shown to influence both user satisfaction and usage of digital health interventions [31]. As it was unclear what the most suitable content would be across both populations, stimuli referring to general health and stimuli referring to eating disorders were used to allow for the most suitable combination of responses to be included in the final analysis.

A greater understanding of these effects was achieved through the development of a mediation model presented later in the paper.

### Primary research question

How do the aesthetics of a digital health intervention impact initial perceptions of the intervention and intentions to use it?

### Secondary research questions

Does perceived trust act as a significant factor in modelling behavioural intentions towards the intervention?

## Materials and methods

### Design

A repeated quasi-experimental online questionnaire study using repeated measures was used, comparing responses to nine different design stimuli across three independent groups. The

nine different design stimuli were an original intervention front page design plus eight alternative designs demonstrating either positive or negative aspects of four aesthetics facets (Simplicity, Diversity, Colour & Craftsmanship). The three independent groups were:

1. People from the general population shown stimuli referring to general health (GP-H).

2. People with experience of an eating disorder shown stimuli referring to eating disorders (ED-ED).

3. People from the general population shown stimuli referring to eating disorders (GP-ED).

The use of a group with eating disorders, as a specific health concern,was intended to extend the study beyond the general population, while considering a population where online resources are frequently accessed and can be of value [32, 33]. Participants in the general population were split into two groups based on stimuli content to assess impacts as a result of keeping the stimuli the same (general population viewing stimuli related to eating disorders (GP-ED)) or adjusting them to increase relevance to the general population (general population viewing stimuli referring to general health (GP-GP)). Including all three groups allowed for analyses to be conducted, as described below, on the most appropriate population sample.

## Participants

Six hundred and one participants with normal, or corrected to normal, vision and no colour blindness were recruited. The general population sample were recruited through various online forums set up to allow members of the public to participate in online research, whilst participants who had experiences of an eating disorder were recruited through newsletters and online forums hosted by UK-based eating disorder charities including Beat Eating Disorders, RestoreED and Men Get Eating Disorders Too, between 8[th] December 2016 and 8[th] December 2017.

Ethical approval was granted by Bournemouth University (ID: 13583) in advance of the study. Consent was collected digitally prior to participants starting the questionnaire using opt-in tick box statements following the digital presentation of the participant information sheet.

## Design stimuli

The design stimuli were created using a novel online digital intervention MotivATE [34, 35]. This intervention was developed to encourage attendance and engagement with eating disorder services, but only the home page of the intervention was used for this study, and the limited material was adapted per user group as required. The four facets of the VisAWI were used to generate static visual stimuli based on the MotivATE intervention's home page. Each of the stimuli were designed to either exemplify (positive stimuli), or fail to exemplify (negative stimuli), each of these visual facets. This resulted in nine stimuli; the original 'Base' version of the home page and a positive and negative version for each of the four facets. In order to separate the aesthetics of the intervention from other factors such as content and usability, which have also been shown to contribute to users' initial impressions [36], the content of each image was kept the same and static, rather than interactive. The nine stimuli, including ratings of each aesthetic facet, are shown in Fig 1. The base and all eight additional stimuli were created, and then rated by 40 individuals, split across two rounds of piloting, on all four aesthetic facets in a pilot test. Details on how each of these facets were implemented in these designs and details of the pilot testing that was conducted to ensure they generated the desired response can be found in S1 Text.

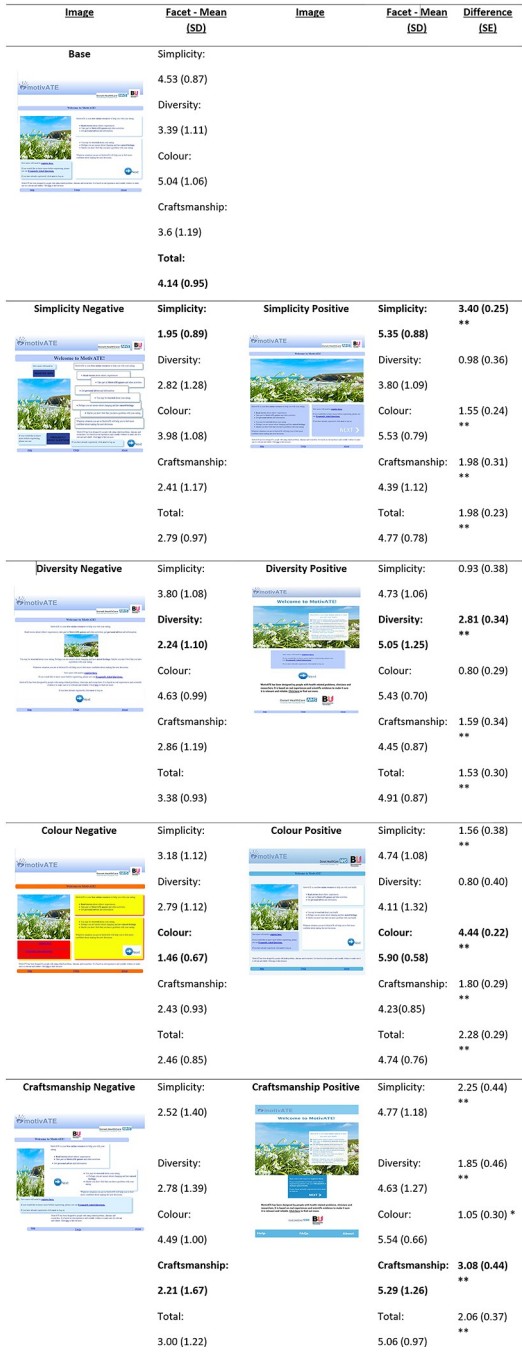

| Image | Facet – Mean (SD) | Image | Facet – Mean (SD) | Difference (SE) |
|---|---|---|---|---|
| **Base** | Simplicity: 4.53 (0.87) | | | |
| | Diversity: 3.39 (1.11) | | | |
| | Colour: 5.04 (1.06) | | | |
| | Craftsmanship: 3.6 (1.19) | | | |
| | **Total: 4.14 (0.95)** | | | |
| **Simplicity Negative** | **Simplicity: 1.95 (0.89)** | **Simplicity Positive** | **Simplicity: 5.35 (0.88)** | **3.40 (0.25) \*\*** |
| | Diversity: 2.82 (1.28) | | Diversity: 3.80 (1.09) | 0.98 (0.36) |
| | Colour: 3.98 (1.08) | | Colour: 5.53 (0.79) | 1.55 (0.24) \*\* |
| | Craftsmanship: 2.41 (1.17) | | Craftsmanship: 4.39 (1.12) | 1.98 (0.31) \*\* |
| | Total: 2.79 (0.97) | | Total: 4.77 (0.78) | 1.98 (0.23) \*\* |
| **Diversity Negative** | Simplicity: 3.80 (1.08) | **Diversity Positive** | Simplicity: 4.73 (1.06) | 0.93 (0.38) |
| | **Diversity: 2.24 (1.10)** | | **Diversity: 5.05 (1.25)** | **2.81 (0.34) \*\*** |
| | Colour: 4.63 (0.99) | | Colour: 5.43 (0.70) | 0.80 (0.29) |
| | Craftsmanship: 2.86 (1.19) | | Craftsmanship: 4.45 (0.87) | 1.59 (0.34) \*\* |
| | Total: 3.38 (0.93) | | Total: 4.91 (0.87) | 1.53 (0.30) \*\* |
| **Colour Negative** | Simplicity: 3.18 (1.12) | **Colour Positive** | Simplicity: 4.74 (1.08) | 1.56 (0.38) \*\* |
| | Diversity: 2.79 (1.12) | | Diversity: 4.11 (1.32) | 0.80 (0.40) |
| | **Colour: 1.46 (0.67)** | | **Colour: 5.90 (0.58)** | **4.44 (0.22) \*\*** |
| | Craftsmanship: 2.43 (0.93) | | Craftsmanship: 4.23 (0.85) | 1.80 (0.29) \*\* |
| | Total: 2.46 (0.85) | | Total: 4.74 (0.76) | 2.28 (0.29) \*\* |
| **Craftsmanship Negative** | Simplicity: 2.52 (1.40) | **Craftsmanship Positive** | Simplicity: 4.77 (1.18) | 2.25 (0.44) \*\* |
| | Diversity: 2.78 (1.39) | | Diversity: 4.63 (1.27) | 1.85 (0.46) \*\* |
| | Colour: 4.49 (1.00) | | Colour: 5.54 (0.66) | 1.05 (0.30) \* |
| | **Craftsmanship: 2.21 (1.67)** | | **Craftsmanship: 5.29 (1.26)** | **3.08 (0.44) \*\*** |
| | Total: 3.00 (1.22) | | Total: 5.06 (0.97) | 2.06 (0.37) \*\* |

**Fig 1. Figure showing the score means (out of a maximum of 7) and SDs of each VisAWI facet for each of the final stimuli designs, as well as the difference and SE between the positive and negative designs for each facet.** *Note*: \*p $< .002$, \*\*p $< .001$. *The target facet for each of the stimuli is highlighted in bold.*

Using these designs second versions of each of the nine stimuli were created by adapting the content referring to eating disorders (as present in the original Motivate intervention) to more widely applicable health related terms. For example, the opening phrase 'MotivATE is your free online course to help you with your eating' was changed to 'MotivATE is your free online course to help you with your health'. This resulted in two sets of stimuli using

identical designs but differing content; one relating to eating disorders (-ED) and one relating to health (-H).

## Measures

Visual aesthetics scores for each of the four design facets were requested from participants using the Short Visual Aesthetics of Websites Inventory (VisAWI-S) [27]. The VisAWI-S was used in place of the full VisAWI to reduce the burden on participants, given the number of stimuli, and was considered appropriate as the stimuli had already been validated using the full VisAWI during the pilot testing.

This study examined seven dependent variables, six of which were derived from the adapted TAM [26] with the addition of trust. The TAM contains seven elements: perceived attractiveness, perceived usefulness, perceived ease of use, perceived enjoyment, attitudes towards use, intention to use and actual usage. The study included outcome measures for all elements of the adapted TAM with the exception of actual usage, as this study aimed to examine only initial impressions and intentions to use, as opposed to actual usage. To address the secondary research questions relating to trust, an additional item to measure this construct was included. Feedback on all items was generated via think aloud interviews, which resulted in some minor alterations to some items to improve clarity.

This resulted in the following seven dependent variables, as assessed by the questions listed in italics, which were measured using responses on 7-point likert scales:

- Perceived attractiveness, *'Overall, I find that the site looks attractive'*, *'The lay-out of the site is attractive'* and *'The colours that are used on the site are attractive'*

- Perceived usefulness, *'I find this to be a useful website'*, *'The information on the site is interesting to me'* and *'I find that this site adds value'*.

- Perceived ease of use, *'I think this is a user-friendly site'*

- Perceived enjoyment, *'I find this website overall an entertaining site'*

- Attitude towards use, *'I have a positive attitude towards this site'*

- Intention to use, *'I intend to visit the site frequently'*, and Intention to register, *'It is likely that I would register with this website'*

- Perceived trust of the intervention, *'I feel that this website is trustworthy'*

Single measures for several of the dependent variables were deemed appropriate as single item measures have been shown to be reliable, sensitive and valid in assessing website constructs [37, 38]. A paper example of the questionnaire can be found in S2 Text.

In addition to these items, the questionnaire also included standard demographic questions (age, gender, nationality), as well as questions relating to previous experience of online self-help, computer literacy and experience of an eating disorder.

## Procedure

Online recruitment material contained links directing participants to the different arms of the trial (ED-ED for those recruited from eating disorder forums/newsletters, and either GP-H or GP-ED for those recruited from other sources). Following collection of online consent, participants were asked to complete the demographic questions listed above, with those indicating that they are currently experiencing an eating disorder being assigned to condition ED-ED whilst those who did not were randomised to conditions GP-H or GP-ED. Following

assignment to a condition, participants were shown some introductory text that framed the questionnaire. Participants were asked to imagine they had been directed to the intervention following referral by their GP. Participants were then asked to complete the 16 measures (4 VisAWI-S subscales and 12 likert scales) for each of the nine stimuli. Each stimulus was present on a separate page with the questionnaire items displayed below. The order in which the stimuli were presented, as well as the question order for each stimulus, were randomised to control for any potential order effects.

On completion of the questionnaire participants were shown a debriefing page outlining the study. In addition to this, the debriefing for condition ED-ED also included links to eating disorder support resources provided by B-EAT.

## Analysis

**The analysis of the data was conducted in two phases.** Phase one focused on exploring the data and preparing an appropriate data set for use in answering the research questions. This included exploring the data for outliers, normality, kurtosis, skewness, and issues of multicollinearity. Previous researchers have suggested that despite the ordinal nature of likert scales, given certain criteria, such as wide enough responses scales (greater than 5), large enough numbers and normal distribution of responses, these data can be treated as quasi-continuous and that parametric tests are robust for the use of such data [39, 40]. As the data in this study fulfilled these criteria, parametric testing was adopted.

As result of concerns regarding multicollinearity, the dependent variables for Attractiveness and Attitude were dropped from the analysis.

Responses between the three groups were then compared at the multivariate level to explore any differences in overall responses to the stimuli and establish which datasets could be combined for inclusion in the primary analysis.

As the analysis of the three population groups was interested in potential differences in responses across all dependent variables for each stimuli, a Multivariate Analysis of Variance (MANOVA) was used. To ensure appropriate testing was used, the assumptions of independence, multivariate normality, and homogeneity of covariance were checked. Mauchly's Test of Sphericity indicated that the assumption of sphericity was violated for each of the dependent variables, as such the Greenhouse-Geiser correction was used. Similarly, Box's M (11080.01) was significant at $p < .001$, suggesting unequal covariance matrices between groups. However, Box's M is known to be highly sensitive to variances in large datasets, with previous researchers suggesting that a MANOVA conducted with greater than 30 participants per condition is robust to violations of this assumption [41]. As such the test was continued but Pillai's trace was reported rather than Wilke's Lamda to ensure the robustness of any conclusions [42]. Due to violations in assumptions of equality of variances, Games-Howell's test was used.

Using these adjustments, a 3x9 mixed MANOVA was used to the explore variance in responses across all three population groups, with follow-up 2x9 mixed MANOVAs to examine differences between each pair of population groups separately. As a result of these tests, the data from the GP-ED group was removed to ensure that any potential variations in responses did not impact on the final models, resulting a in a final dataset from the combined data from groups GP-H and ED-ED.

The second phase of the analysis sought to answer the primary and secondary research questions using this dataset. Linear mixed models were used to explore interactions between the dependent variables. A penalized likelihood approach was used to find the best model fit in regards to covariance structure and random effects with a smaller Bayesian Information Criterion (BIC) score difference greater than two being used to suggest a more accurate model fit to the data [43].

The appropriate fit of random intercept and slope models including participant and design as random effects was explored, using AR1, Diagonal, Compound Symmetry and Huynh-Feldt covariance structures. Each model produced a similar pattern of results, with the BIC indicating that a random slope model for design plus intercept using an AR1 covariance structure and design repeated statement offered the best model fit. Final models were run using both restricted maximum likelihood (REML) and maximum likelihood (ML) estimations, which resulted in matching patterns of results and very similar fixed effect estimates. As such ML models are shown for ease of comparison between models with differing fixed effects [44].

Four models were generated to investigate the best predictive model of behavioural intentions:

- The first model examined the direct influence of simplicity, diversity, colour and craftsmanship on behavioural intentions.

- The second model added the TAM factors of ease of use, perceived enjoyment and perceived usefulness as covariates.

- The third model added the final variable of trust as an additional covariate.

- A final, simplified, model was then produced including only the significant predictors from model three.

As each model contained several fixed effects, Bonferoni corrected critical p-values ($p < = .006$) were used in each case to assess whether predictors made a significant contribution to the model.

Next, a more detailed explanation of the observed impact on behavioural intentions of the mediating effects of the final model were explored. A structural equation modelling approach was considered in order to develop a full path model, however the nature of our data, due to the use of single items as well as the high level of complexity that would be introduced to such a model as a result of repeated measures, meant that a simpler approach to visualising mediation was adopted using repeated mixed models. In each case random effects and covariance structure were standardised, and a ML estimation was used, in order to ensure that fixed effects coefficients were comparable.

In this instance four further models were generated, examining the fixed effects of the significant VisAWI facets from the final predictive model (Simplicity and Craftsmanship) on the proposed mediating factors of Usefulness (model A), Ease of Use (model B), Enjoyment (model C) and Trust (model D), which were then combined with the fixed effects from the final model (model E) to create a visual representation of the proposed mediation pathways.

All analyses were conducted using the IBM SPSS statistics package 23.

## Results

### Phase one

In total, 601 participants were recruited to the study. Full details of participant characteristics, both as a full sample and for each population group, are shown in Table 1.

Normality was checked for each of the recorded variables, both collectively and for each condition, with Kolomogorov-Smirnov statistics suggesting that data were non-normal. However, in each instance Q-Q plots closely matched the expected normal, and kurtosis and skewness were found to be well within acceptable limits of ±2, suggesting an acceptable level of normality in the data despite the indicated test statistic [45, 46]. Twenty-four outliers were identified, which were further explored in line with recommendations by Aguinis, Gottfredson and Joo [47]. As all data were collected digitally it is unlikely that outliers resulted from

**Table 1. Participant characteristics.**

| | GP-H | ED-ED | GP-ED | Total |
|---|---|---|---|---|
| | N = 257 | N = 109 | N = 235 | N = 601 |
| Age | 18–24: 174 | 18–24: 69 | 18–24: 163 | 18–24: 406 |
| | 25–34: 60 | 25–34: 35 | 25–34: 51 | 25–34: 146 |
| | 35–44: 14 | 35–44: 4 | 35–44: 12 | 35–44: 30 |
| | 45–54: 6 | 45–54: 0 | 45–54: 5 | 45–54: 11 |
| | 55–64: 3 | 55–64: 1 | 55–64: 1 | 55–64: 5 |
| | 65–74: 0 | 65–74: 0 | 65–74: 2 | 65–74: 2 |
| | 75–84: 0 | 75–84: 0 | 75–84: 0 | 75–84: 0 |
| | 85 or older: 0 | 85 or older: 0 | 85 or older: 0 | 85 or older: 0 |
| | | | No response: 1 | No response: 1 |
| Gender | Male: 99 | Male: 17 | Male: 82 | Male: 198 |
| | Female: 149 | Female: 86 | Female: 144 | Female: 379 |
| | Other: 2 | Other: 4 | Other: 2 | Other: 8 |
| | Prefer Not To Say: 1 | Prefer Not To Say: 2 | Prefer Not To Say: 1 | Prefer Not To Say: 4 |
| Accessed Online Advice | Yes: 205 | Yes: 88 | Yes: 194 | Yes: 487 |
| | No: 52 | No: 21 | No: 41 | No: 114 |
| Accessed Online Self-Help | Yes: 128 | Yes: 61 | Yes: 109 | Yes: 298 |
| | No: 129 | No: 48 | No: 126 | No: 303 |
| Previously Had An Eating Disorder | Yes: 24 | N/A | Yes: 24 | Yes: 48 |
| | No: 233 | | No: 211 | No: 444 |
| | | | | N/A: 109 |
| Attended An Eating Disorder Service | Yes: 6 | Yes: 54 | Yes: 9 | Yes: 69 |
| | No: 18 | No: 55 | No: 15 | No: 88 |
| | N/A: 233 | | N/A: 211 | N/A: 444 |
| Self Reported Eating Disorder (past or present) | Anorexia Nervosa: 8 | Anorexia Nervosa: 44 | Anorexia Nervosa: 11 | Anorexia Nervosa: 63 |
| | Bulimia Nervosa: 6 | Bulimia Nervosa: 22 | Bulimia Nervosa: 5 | Bulimia Nervosa: 33 |
| | Binge Eating Disorder: 2 | Binge Eating Disorder: 13 | Binge Eating Disorder: 1 | Binge Eating Disorder: 16 |
| | EDNOS/OSFED: 3 | EDNOS/OSFED: 17 | EDNOS/OSFED: 2 | EDNOS/OSFED: 22 |
| | Other: 5 | Other: 10 | Other: 4 | Other: 19 |
| | N/A: 233 | No response: 3 | N/A: 212 | No response: 3 |
| | | | | N/A: 445 |
| From The UK | Yes: 188 | Yes: 93 | Yes: 172 | Yes: 453 |
| | No: 69 | No: 15 | No: 61 | No: 145 |
| | Missing: 0 | Missing: 1 | Missing: 2 | Missing: 3 |
| English As First Language | Yes: 214 | Yes: 93 | Yes: 193 | Yes: 500 |
| | No: 43 | No: 16 | No: 40 | No: 99 |
| | | | No response: 2 | No response: 2 |
| Computer Use | Once a month or less: 0 | Once a month or less: 0 | Once a month or less: 0 | Once a month or less: 0 |
| | Once a week: 1 | Once a week: 0 | Once a week: 0 | Once a week: 2 |
| | Several times a week: 13 | Several times a week: 4 | Several times a week: 12 | Several times a week: 29 |
| | Every day: 96 | Every day: 59 | Every day: 81 | Every day: 236 |
| | Several times a day: 147 | Several times a day: 46 | Several times a day: 141 | Several times a day: 334 |

(*Continued*)

**Table 1.** (Continued)

|  | GP-H | ED-ED | GP-ED | Total |
|---|---|---|---|---|
|  | N = 257 | N = 109 | N = 235 | N = 601 |
| Computer Time Per Week | 0 to 1 hours: 1 | 0 to 1 hours: 0 | 0 to 1 hours: 0 | 0 to 1 hours: 1 |
|  | 2 to 4 hours: 14 | 2 to 4 hours: 5 | 2 to 4 hours: 15 | 2 to 4 hours: 34 |
|  | 5 to 6 hours: 19 | 5 to 6 hours: 6 | 5 to 6 hours: 15 | 5 to 6 hours: 40 |
|  | 7 to 9 hours: 21 | 7 to 9 hours: 8 | 7 to 9 hours: 19 | 7 to 9 hours: 48 |
|  | 10 to 20 hours: 48 | 10 to 20 hours: 28 | 10 to 20 hours: 41 | 10 to 20 hours: 117 |
|  | 21 to 40 hours: 90 | 21 to 40 hours: 40 | 21 to 40 hours: 75 | 21 to 40 hours: 205 |
|  | Over 40 hours: 64 | Over 40 hours: 22 | Over 40 hours: 69 | Over 40 hours: 155 |
|  |  |  | No response: 1 | No response: 1 |

EDNOS: Eating Disorder Not Otherwise Specified, OSFED: Otherwise Specified Eating Disorder

incorrectly inputted data, and as demographic information for these outliers was similar to the general data set and reasonably distributed across groups, this was not deemed sufficient reason to remove what could potentially be meaningful and interesting data. All results reported were therefore derived from data sets including these recorded outliers.

Both the measures of Attractiveness and Attitude were shown to have high levels of collinearity with several other variables. These checks resulted in the removal of the measure of Attitude due to concerns regarding multicollinearity (Tolerance = .181; VIF = 5.537), whilst the measure of Attractiveness was shown to correlate strongly with the compound VisAWI scores (r = 0.945, n = 5409, $p < .001$) and as such was dropped in favour of the more detailed approach of using the four facets of the VisAWI individually. A correlation table is provided in S1 Table. Of the five remaining dependent variables explored in this study, two (usefulness and behavioural intentions) were examined using multiple measures. Reliability analyses were conducted for these variables, resulting in Cronbach's Alphas of .852 and .913 respectively. This represents a high level of reliability for each set of measures, and as such a single compound measure was generated for each dependent variable for use in the final analysis. Results from the MANOVA indicated that there were no statistically significant differences in participants' responses across all three participant groups ($F$ (18, 1182) = 1.563, $p = .062$; Pillai's Trace = 0.047, partial $\eta2 = .02$). The analysis did however highlight a significant interaction effect for responses between stimuli for each group ($F$ (144, 10256) = 1.309, $p < .05$; Pillai's Trace = 0.303, partial $\eta2 = .15$). This would suggest that the way in which responses varied between stimuli was different between the three groups.

Follow-up analyses found significant multivariate group differences between groups ED-ED and GP-ED ($F$ (9, 334) = 2.208, $p < .01$; Pillai's Trace = 0.056, partial $\eta2 = .06$) but no stimuli by group interaction. However no significant differences were found between groups ED-ED and GP-H or GP-ED and GP-H for either group or stimuli by group effects. As such final models were produced from pooled responses from groups GP-H and ED-ED, those who viewed stimuli relevant to their health status (N = 366), with the data from the GP-ED group being removed to ensure that any potential variations in responses did not impact on the final models.

## Phase two

Using this combined dataset, four linear mixed models were generated to explore the role of aesthetics on participants' behavioural intentions.

The results of the four models are shown in Table 2.

**Table 2.** *Linear mixed models for predictors of user behavioural intentions.*

| Model | | Parameter | Estimate | Std. Error | Sig. | 95% Confidence Interval | |
|---|---|---|---|---|---|---|---|
| | | | | | | Lower Bound | Upper Bound |
| One | Fixed Effects | Intercept | .577 | .048 | < .001 | .482 | .671 |
| | | Simplicity | .237 | .013 | . < .001 | .212 | .262 |
| | | Diversity | .118 | .012 | < .001 | .094 | .141 |
| | | Colour | .092 | .011 | < .001 | .071 | .113 |
| | | Craftsmanship | .293 | .013 | < .001 | .267 | .319 |
| | Random Effects | Repeated | .125 | .032 | < .001 | .076 | .207 |
| | | Intercept + Stimuli [subject = Participant] | .319 | .029 | < .001 | .266 | .382 |
| | Model Information Criteria | AIC | 7443.926 | | | | |
| | | BIC | 7492.725 | | | | |
| Two | Fixed Effects | Intercept | -.218 | .047 | < .001 | -.310 | -.126 |
| | | Simplicity | .110 | .013 | < .001 | .085 | .134 |
| | | Diversity | .020 | .011 | .065 | -.001 | .042 |
| | | Colour | .038 | .010 | < .001 | .019 | .058 |
| | | Craftsmanship | .191 | .012 | < .001 | .168 | .215 |
| | | Ease Of Use | .092 | .013 | < .001 | .067 | .117 |
| | | Usefulness | .339 | .016 | < .001 | .306 | .371 |
| | | Enjoyment | .157 | .013 | < .001 | .131 | .183 |
| | Random Effects | Repeated | .166 | .019 | < .001 | .133 | .208 |
| | | Intercept + Stimuli [subject = Participant] | .182 | .016 | < .001 | .152 | .216 |
| | Model Information Criteria | AIC | 6552.662 | | | | |
| | | BIC | 6619.760 | | | | |
| Three | Fixed Effects | Intercept | -.242 | .046 | < .001 | -.333 | -.151 |
| | | Simplicity | .092 | .013 | < .001 | .067 | .117 |
| | | Diversity | .016 | .011 | .125 | -.005 | .038 |
| | | Colour | .024 | .010 | .014 | .005 | .043 |
| | | Craftsmanship | .158 | .012 | < .001 | .134 | .182 |
| | | Ease Of Use | .059 | .013 | < .001 | .033 | .084 |
| | | Usefulness | .293 | .017 | < .001 | .260 | .325 |
| | | Enjoyment | .158 | .013 | < .001 | .133 | .184 |
| | | Trust | .153 | .014 | < .001 | .126 | .180 |
| | Random Effects | Repeated | .148 | .019 | < .001 | .115 | .190 |
| | | Intercept + Stimuli [subject = Participant] | .185 | .016 | < .001 | .156 | .220 |
| | Model Information Criteria | AIC | 6432.774 | | | | |
| | | BIC | 6505.973 | | | | |
| Four | Fixed Effects | Intercept | -.217 | .045 | < .001 | -.307 | -.128 |
| | | Simplicity | .100 | .012 | < .001 | .076 | .124 |
| | | Craftsmanship | .168 | .012 | < .001 | .145 | .190 |
| | | Ease Of Use | .058 | .013 | < .001 | .032 | .083 |
| | | Usefulness | .300 | .017 | < .001 | .267 | .332 |
| | | Enjoyment | .166 | .013 | < .001 | .141 | .190 |
| | | Trust | .159 | .014 | < .001 | .132 | .185 |
| | Random Effects | Repeated | .150 | .019 | < .001 | .117 | .192 |
| | | Intercept + Stimuli [subject = Participant] | .184 | .016 | < .001 | .155 | .219 |
| | Model Information Criteria | AIC | 6437.921 | | | | |
| | | BIC | 6498.919 | | | | |

**Table 3.** *Linear mixed models for each element in the mediation diagram.*

| Model (Dependent Variable) | | Parameter | Estimate | Std. Error | Sig. | 95% Confidence Interval | |
|---|---|---|---|---|---|---|---|
| | | | | | | Lower Bound | Upper Bound |
| A (Usefulness) | Fixed Effects | Intercept | 1.935 | .047 | < .001 | 1.841 | 2.028 |
| | | Simplicity | .263 | .013 | < .001 | .238 | .288 |
| | | Craftsmanship | .272 | .012 | < .001 | .248 | .296 |
| | Random Effects | Repeated | .491 | .013 | < .001 | .467 | .517 |
| | | Intercept + Stimuli [subject = Participant] | .400 | .033 | < .001 | .340 | .470 |
| B (Ease Of Use) | Fixed Effects | Intercept | 1.073 | .045 | < .001 | .984 | 1.161 |
| | | Simplicity | .486 | .015 | < .001 | .456 | .516 |
| | | Craftsmanship | .310 | .015 | < .001 | .281 | .338 |
| | Random Effects | Repeated | .766 | .020 | < .001 | .727 | .806 |
| | | Intercept + Stimuli [subject = Participant] | .150 | .017 | < .001 | .120 | .188 |
| C (Enjoyment) | Fixed Effects | Intercept | 1.463 | .053 | < .001 | 1.359 | 1.567 |
| | | Simplicity | .208 | .016 | < .001 | .177 | .239 |
| | | Craftsmanship | .317 | .015 | < .001 | .287 | .346 |
| | Random Effects | Repeated | .767 | .020 | < .001 | .728 | .807 |
| | | Intercept + Stimuli [subject = Participant] | .387 | .034 | < .001 | .326 | .460 |
| D (Trust) | Fixed Effects | Intercept | 1.126 | .047 | < .001 | 1.034 | 1.217 |
| | | Simplicity | .353 | .015 | < .001 | .324 | .382 |
| | | Craftsmanship | .401 | .014 | < .001 | .373 | .429 |
| | Random Effects | Repeated | .693 | .018 | < .001 | .658 | .729 |
| | | Intercept + Stimuli [subject = Participant] | .238 | .023 | < .001 | .197 | .288 |
| E (Behavioural Intentions) | Fixed Effects | Intercept | -.217 | .045 | < .001 | -.307 | -.128 |
| | | Simplicity | .100 | .012 | < .001 | .076 | .124 |
| | | Craftsmanship | .168 | .012 | < .001 | .145 | .190 |
| | | Ease Of Use | .058 | .013 | < .001 | .032 | .083 |
| | | Usefulness | .300 | .017 | < .001 | .267 | .332 |
| | | Enjoyment | .166 | .013 | < .001 | .141 | .190 |
| | | Trust | .159 | .014 | < .001 | .132 | .185 |
| | Random Effects | Repeated | .150 | .019 | < .001 | .117 | .192 |
| | | Intercept + Stimuli [subject = Participant] | .184 | .016 | < .001 | .155 | .219 |

As can be seen from Table 3 whilst each of the aesthetics measures do significantly predict behavioural intentions to use the intervention when considered by themselves, inclusion of the TAM measures, results in no significance effect for diversity, though all other measures remain significant. The addition of trust in model three further reduces the effect of diversity, as well as reducing the effects of colour, though all other measures remain significant. In addition to this, it can be seen that including the TAM measures in the model, as well as additionally adding the measure of trust, presents a better model fit. Finally, removing the non-significant (when using a Bonferoni corrected critical p-value of 0.006) measures of diversity and colour from the model appears not only to have no negative effect on the model fit as shown by the AIC, but when considering the BIC actually gives a better fit than when they are included. Additionally, in each model the random effects are significant, suggesting that correlations between participants responses to each of the stimuli accounted for a significant portion of the variation in each model. The relationships between each dependent variable included in model four were then explored using further linear mixed models, resulting in five further models for the dependent variables Usefulness (model A), Ease of Use (model B), Enjoyment

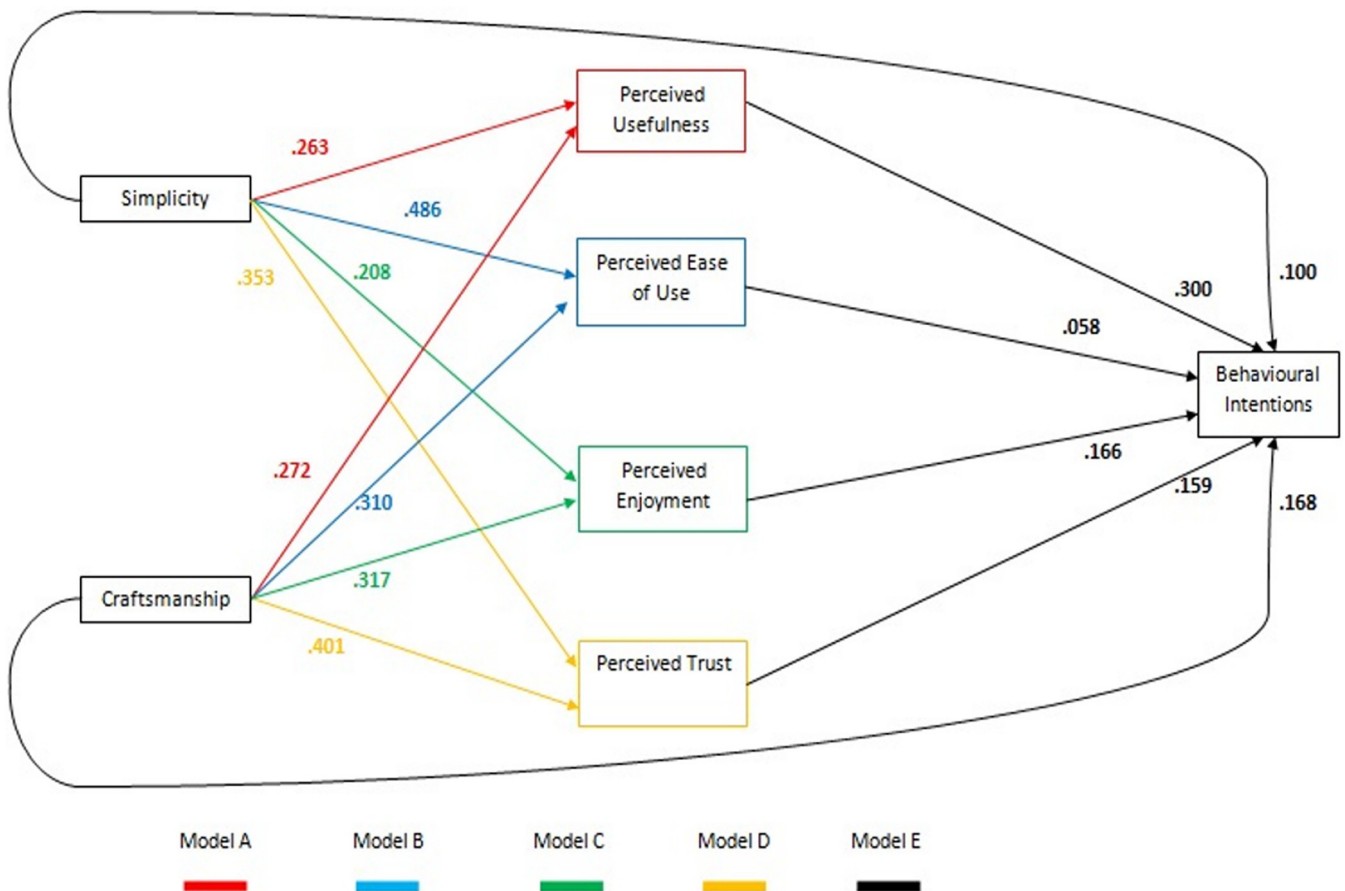

**Fig 2. Mediation diagram of effects on behavioural intentions, colour coded by the linear mixed model conducted on each dependent variable as detailed in Table 3.** Coefficient estimates are derived from the individual models (A-E).

(model C), Trust (model E) and Behavioural Intentions (model E). The outputs of these analysis are shown in Table 3.

The resulting diagram shown in Fig 2 therefore represents a 'collage' of models presented in Table 3, mapping the fixed effects for the relationships between each variable and colour coded in accordance to models A-E. This has been done in order to give a clearer depiction of the relationship between the significant model factors, but does not constitute a model in and of itself. As such no model fit statistics are reported.

Fig 2 shows the resulting mediation diagram based on the TAM model proposed by Van der Heijden [26], with the addition of 'perceived trust', and visual attractiveness broken down into the two facets found to have a significant effect on behavioural intentions (Simplicity and Craftsmanship). Attitude is not included in the diagram as it was removed earlier in the analysis as a result of issues with multi-collinearity. Similarly, the VisAWI facets of Colour and Diversity are not included as they were shown to have a non-significant impact on behavioural intentions.

As seen from model E, both the design elements of Simplicity and Craftsmanship had modest yet significant direct effects (.100 and .168 respectively) on user Behavioural Intentions, with the proposed mediator variables of Perceived Enjoyment and Perceived Trust having similar direct effects (.166 and .159 respectively). The proposed mediator or Perceived Usefulness

had the strongest impact on Behavioural Intentions, with a direct effect of .300, whilst Perceived Ease of Use had the weakest, with an effect of .058.

Judgments of Perceived Usefulness (model A) were influenced similarly by both Simplicity and Craftsmanship, with fixed effects of .263 and .267 respectively. Perceived Ease of Use (model B) was strongly influenced by the Simplicity of the designs (.486), and to a slightly lesser extent their Craftsmanship (.310). Perceived Enjoyment (model C) showed the weakest mediating effect from Simplicity (.208) along with a slightly stronger effect from Craftsmanship (.317). Finally, Perceived Trust (model D) provided a significant addition to the model, being influenced by judgements of Simplicity (.353) and to a greater degree Craftsmanship (.401).

## Discussion

This research provides evidence to support a role for aesthetic design in promoting positive user judgements of online interventions, in particular, for the roles of simplicity and craftsmanship. Design simplicity was seen to have a particular influence on judgements of perceived ease of use, indicating that simpler designs are more accessible to users. Craftsmanship was found to have an influence on both perceived trust and perceived ease of use of the intervention, suggesting that the perception of greater care and skill in the design not only makes it seem easier to use but also heightens the user's sense that the intervention is reliable and of high quality.

The model generated by this research also supports the inclusion of perceived trust as a predictor of behavioural intentions towards digital health interventions. This highlights the importance of building user trust, in addition to using positive aesthetic features in all aspects of the design process. Indeed, regarding the provision of online health information, the quality of content has been shown to strongly influence patient trust [48].

These findings support the application of the adapted TAM framework put forward by Van der Heijden [26] to digital health interventions, as well as the addition of the role of perceived trust. This provides an effective structure to facilitate future understanding of how users perceive and engage with digital health interventions. More specifically the relationship between simplicity and perceived ease of use identified by Lazard et al. [20] in relation to online patient portals was also supported by this research, suggesting that this may be a widely applicable pathway for improving user experiences of digital health interventions.

It is worth noting however, that the actual effect sizes of the aesthetic, TAM and trust facets, while statistically significant, were relatively small. This would suggest that whilst influencing user behavioural intentions towards digital health interventions to some degree, these are only several of many factors that must come together to create an intervention that users are most likely to engage with.

Some differences were found in the responses between individuals with a specific health condition (eating disorders) and the general population. However, these differences were only present at the multivariate level, indicating that this observation was not a result of differences in responses to any particular variable, but instead represented a complex variation in responses across all elements of the questionnaire. This makes it difficult to assess exactly how the two populations differ in their judgements of digital health interventions. Furthermore, follow-up analysis indicated that these differences were only present when the two populations were responding to stimuli that related to eating disorders (between conditions ED-ED and GP-ED). This may suggest that differences in responses may have been due to the relevance of the content in the stimuli rather than as a result of any quality or characteristic relating to the individual. In other words, there were no differences in the responses of people with eating disorders and people from the general population when the stimuli were relevant to them

(referring to eating disorders or health for each group respectively), but responses did differ between these populations when both were shown stimuli that referred to eating disorders (relevant to one group but not the other). This serves to further highlight the importance not only of the design of the intervention, but of the content that it provides, and in particular the relevance that this content has to the target population. However, it is worth noting that this research was not specifically designed to investigate the role of content on user judgements, and as such further investigations specifically of relevance may be of value.

A key strength of this research was the controlled nature of the design stimuli. By using specifically designed stimuli that covered a range of design features, as opposed to previous research that relied primarily on examples taken from websites in current usage, the impacts of a wider range of user responses were systematically explored. This allowed for a clearer picture of the role that different aesthetic principles play in the development of user judgements towards digital interventions. Interestingly, diversity and colour were found to have very limited impacts in this respect. The inclusion of different populations, as well as comparisons of different content relating to these populations, also allowed for a more subtle exploration of both the wider applicability of the model generated by this research as well as the potential interplay between design and content. The large sample size, and the use of a within-subjects design to assess the different design stimuli also created a large data set.

However, this research did suffer from a number of limitations. Perhaps the most notable of these was the nature of the questionnaire used to record participant responses. Whilst the use of largely single item measures is supported as a viable approach to assessing user attitudes towards websites, and was chosen in order to reduce the cognitive load on participants as a result of the repeated-measures design, the nature of these data did prohibit more in-depth structural analysis of the model. Future research using more robust measures to allow for a full structural equation modelling approach of the data to confirm the mediated model is recommended. In addition, future research would potentially benefit from the involvement of professional website designers in the development of the stimuli. Whilst the stimuli used in this study did successfully generate a range of responses, the positive stimuli failed to access the highest responses (out of the possible maximum score of 7) from participants, suggesting that these could be improved. Further research to identify exactly what aspects users perceived to exemplify the highest levels of each aesthetic facet would be of interest.

Whilst this study provides initial evidence for a model of the role of aesthetics in improving digital health interventions, further research is needed in a number of directions in order to both confirm these findings and improve their usefulness to clinicians and developers. Firstly, this research only links the role of aesthetics to improving users' behavioural intentions towards the intervention. Whilst the relationship between behavioural intentions and actual behaviour has been established [49], further research demonstrating actual differences in user engagement with digital health interventions as a result of changes in the aesthetic design is required in order to fully establish this effect. Furthermore, as mentioned above, while our model identifies simplicity and craftsmanship as important aspects of design for digital interventions, we can provide no specific information as to how positive judgements of these facets can be achieved. As such further research is also required in order to ascertain exactly what designs constitute positive implementation of each of the key aesthetic facets so that practical recommendations can be made to future developers.

## Conclusions

In conclusion, this study isolated a number of digital intervention design features, and tested a role for these in future intentions. The findings provide a number of novel and clinically

relevant insights. Firstly, further developing the TAM model proposed by Van der Heijden [26] allows for a more nuanced insight into the role of visual aesthetics on behavioural intentions, in particular the influence of Simplicity and Craftsmanship, as well as how these factors interact with more established elements of the model. Secondly, the addition of perceived trust to the model builds on the potential importance of this factor as highlighted by previous research and identifies it as an important consideration in the development of digital intervention.

## Supporting information

**S1 Text. Overview of pilot study work.**
(DOCX)

**S2 Text. Paper version of questionnaire.**
(DOCX)

**S1 Table. Correlations between study variables.**
(XLSX)

## Acknowledgments

This work was undertaken as part of a PhD being completed by JDD, under the supervision of SM, KMA, and CN.

## Author Contributions

**Conceptualization:** James L. Denison-Day, Sarah Muir, Katherine M. Appleton.

**Data curation:** James L. Denison-Day.

**Formal analysis:** James L. Denison-Day.

**Investigation:** James L. Denison-Day.

**Methodology:** James L. Denison-Day.

**Project administration:** James L. Denison-Day.

**Supervision:** Sarah Muir, Ciaran Newell, Katherine M. Appleton.

**Writing – original draft:** James L. Denison-Day.

**Writing – review & editing:** Sarah Muir, Ciaran Newell, Katherine M. Appleton.

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
