## [Decision Letter · Decision Letter 0]

26 Jan 2023

PDIG-D-22-00299

The Role of Aesthetics in Intentions to Use Digital Health Interventions

PLOS Digital Health

Dear Dr. Denison-Day,

Thank you for submitting your manuscript to PLOS Digital Health. After careful consideration, we feel that it has merit but does not fully meet PLOS Digital Health's publication criteria as it currently stands. Therefore, we invite you to submit a revised version of the manuscript that addresses the points raised during the review process.

Specifically, I would like to encourage you to critically revise the description of methods and results to improve the clarity of the manuscript. One reviewer furthermore provided comments on the data analysis approach that you might want to take into account.

Please submit your revised manuscript within 60 days Mar 27 2023 11:59PM. If you will need more time than this to complete your revisions, please reply to this message or contact the journal office at digitalhealth@plos.org. Please include the following items when submitting your revised manuscript:

We look forward to receiving your revised manuscript.

Kind regards,

Laura M. König

Academic Editor

PLOS Digital Health

Journal Requirements:

Additional Editor Comments (if provided):

Please carefully consider all reviewer comments - Reviewer 1 in particular had specific concerns regarding the methods and analyses used. These concerns must be addressed in any resubmission. 

Reviewers' comments:

Reviewer's Responses to Questions

**Comments to the Author**

1. Does this manuscript meet PLOS Digital Health’s publication criteria? Is the manuscript technically sound, and do the data support the conclusions? The manuscript must describe methodologically and ethically rigorous research with conclusions that are appropriately drawn based on the data presented.

Reviewer #1: No

Reviewer #2: Yes

2. Has the statistical analysis been performed appropriately and rigorously?

Reviewer #1: I don't know

Reviewer #2: I don't know

3. Have the authors made all data underlying the findings in their manuscript fully available (please refer to the Data Availability Statement at the start of the manuscript PDF file)?

Reviewer #1: Yes

Reviewer #2: Yes

4. Is the manuscript presented in an intelligible fashion and written in standard English?

Reviewer #1: Yes

Reviewer #2: Yes

5. Review Comments to the Author

Reviewer #1: Thank you for the opportunity to review your work in, "The Role of Aesthetics in Intentions to Use Digital Health Interventions." I appreciate the need for this research and very much agree with the authors that evidence-based approaches for designing digital health interventions are necessary to increase the likelihood of adoption. This work also appears to have a potentially informative study design (albeit different than described) and rigorously crafted stimuli that set the authors up well to contribute to the evidence base in this area. However, as written, the methods, in particular the measures and analyses, and findings are not clear, thus, reducing the transparency and potential impact of the work. Below are some comments for consideration. 

Abstract 

The abstract is the gateway to your work having an impact, and it is clear you have put the time and energy into making it accessible and informative. Very nice! My only minor suggestion is to make explicit the experimental design is within subjects with a repetition among three different populations (see note in methods about this clarification too). 

Introduction 

Overall the introduction is well written, making a clear case for the research. I wish the authors had gone with hypotheses instead of research questions, but alas, it it too late for that. Something to consider in future work. 

One other thought. Consider cutting the TAM figure. It doesn't add much, and doesn't represent current TAM models (e.g., TAM2, UTAUT). The written argument still holds, as these are still the central pathways of the newer models, but the figure could be removed, which also allows room for more important information (see note below about stimuli). 

Methods

Although this study is described as a "mixed-design," presumably with a 3 level between-persons factors and 9 level within-persons factors, it is not truly a mixed design. Here your between-persons factor isn't a randomized factor, but rather a repetition of the experiment -- albeit with different stimuli? -- among three populations. Please revise to clarify to ensure the study design is not misread. It is a small clarification in words, but has tremendous impact for how the work can be interpreted. You essentially have three different within-persons studies. This is a valuable design, even if not a mixed design. 

I highly recommend having your final stimuli in the S1 Appendix as a main table. It is fine to keep the process as an Appendix. But without this main table with stimuli, the study is hard to understand. The stimuli is essential for readers. Cut other information to make room (e.g., TAM figure in intro doesn't add much). 

It is still not clear how the stimuli content differed for the three populations, the repetition in this research. Please update with more information. This is a critical addition. 

Please include more detail for the number of items for each construct in the methods. This is needed for transparency, potential repeatability of the study, and interpretation of the data. It is not clear what was asked of the participants. 

As written, how content relevance could be addressed as a between-persons comparison is not clear. Were participants randomized to see content? It doesn't appear so. 

The subgroup analyses are first introduced in the data analyses section. These are not well justified. Please address. 

Most importantly, the data analysis plan is missing a clear account of what variables were used in what analyses. A MANOVA is not well justified. I do not recommend this approach. In the revisions, I encourage the authors to consider a revised plan that better aligns with the study design. Despite the choices made here, there needs to be a clear account of all predictor and outcome variables for each test conducted. 

Some tests are described in the methods and some in the results. While all findings should go in the results, a clear description with all tests conducted should be in one place -- preferably in the data analysis section of the methods. 

Two phases are described in the data analyses, but more appear in the result. Please revise. None of the models presented in the results are justified or described up to this point. 

Results

Given my confusion about the analyses (see note above), I am struggling to follow the results. Beyond this, the results are generally referred to by the test conducted and not the variables making it even harder to follow. Please revise to also put the variables with whether the findings are significance or not. 

The abbreviations in the results are tremendously hard to follow in the text. These might work ok for the tables where a key can be provided direct below, but in text it is near impossible to extract meaning without the variable written out. 

Significance tests produce a single static number. There can be no trending or other action verb for these results. Please remove. If something is not significant, and by your planned adjusted p-value this would be <=.006, then it is not significant. That doesn't make a finding any less important. Reporting without an action verb does is critical to avoid misleading readers. 

There is inconsistency in the reporting. Some findings were reported as significant at p <.05, which doesn't take into account the adjusted value set up in the methods. Please revise. 

While I was potentially still hanging on Table 4, a series of models looking at a single mediator between aesthetics and behavioral intentions that were newly introduced and not justified, I am completely lost with Table 5. I highly recommend removing these analyses in Table 5, as they do not appear to follow any recommended guidelines for analysis of a within-persons experiment nor are they justified. 

Discussion

Unfortunately, because I became so lost in the methods and results, I am not able to review the discussion. I am unable to interpret the findings to determine whether the discussion aligns with the data or not. 

Data

Please update the link to the data. There is currently an error making it so one can't access the data. I did select "yes" above though, as I believe this is an error and not intentional.

Reviewer #2: This is a well written study that appears to be rigorously conducted. I don't have expertise in the statistical methods so cannot comment on them but my impression was that the statistical analysis was described well, with justification when required, and presented clearly. 

The rationale and methods are clear. 

The results are hard to follow in parts due to the complexity of the statistics and the diagram with the Models (A-E) needs a little more description in the figure heading or footnote. 

The Discussion is appropriate.

The findings are relevant in terms of advancing the field of digital health. 

I have no further feedback or edits suggested

6. PLOS authors have the option to publish the peer review history of their article (what does this mean?). If published, this will include your full peer review and any attached files.

**Do you want your identity to be public for this peer review?** For information about this choice, including consent withdrawal, please see our Privacy Policy.

Reviewer #1: No

Reviewer #2: No

---

## [Decision Letter · Decision Letter 1]

16 May 2023

The Role of Aesthetics in Intentions to Use Digital Health Interventions

PDIG-D-22-00299R1

Dear Dr Denison-Day,

We are pleased to inform you that your manuscript 'The Role of Aesthetics in Intentions to Use Digital Health Interventions' has been provisionally accepted for publication in PLOS Digital Health.

Best regards,

Laura M. König

Academic Editor

PLOS Digital Health

Reviewer Comments (if any, and for reference):

Reviewer's Responses to Questions

**Comments to the Author**

1. If the authors have adequately addressed your comments raised in a previous round of review and you feel that this manuscript is now acceptable for publication, you may indicate that here to bypass the “Comments to the Author” section, enter your conflict of interest statement in the “Confidential to Editor” section, and submit your "Accept" recommendation.

Reviewer #1: All comments have been addressed

Reviewer #2: All comments have been addressed

2. Does this manuscript meet PLOS Digital Health’s publication criteria? Is the manuscript technically sound, and do the data support the conclusions? The manuscript must describe methodologically and ethically rigorous research with conclusions that are appropriately drawn based on the data presented.

Reviewer #1: Yes

Reviewer #2: Yes

3. Has the statistical analysis been performed appropriately and rigorously?

Reviewer #1: Yes

Reviewer #2: I don't know

4. Have the authors made all data underlying the findings in their manuscript fully available (please refer to the Data Availability Statement at the start of the manuscript PDF file)?

Reviewer #1: Yes

Reviewer #2: Yes

5. Is the manuscript presented in an intelligible fashion and written in standard English?

Reviewer #1: Yes

Reviewer #2: Yes

6. Review Comments to the Author

Reviewer #1: Thank you for these thoughtful revisions. Many of my concerns have been addressed. This is a thoughtfully designed study with clear implications.

One request prior to publishing: Remove all verb language for statistical significance. P-values are static numbers and can't have an action. There is no "trending" or other movement

Reviewer #2: I have no additional comments

7. PLOS authors have the option to publish the peer review history of their article (what does this mean?). If published, this will include your full peer review and any attached files.

**Do you want your identity to be public for this peer review?** For information about this choice, including consent withdrawal, please see our Privacy Policy.

Reviewer #1: No

Reviewer #2: No
